# Research on the Impact of Environmental Regulations on Industrial Green Total Factor Productivity: Perspectives on the Changes in the Allocation Ratio of Factors among Different Industries

**Jiaqi Yuan [1],\* and Deyuan Zhang [2]**

1   School of Economics and Management, Beijing Jiaotong University, Beijing 100044, China
2   Institute of Economic System and Management, Academy of Macroeconomic Research,
    National Development and Reform Commission, Beijing 100035, China; zhangdy271@126.com
\*   Correspondence: yuanjiaqi@bjtu.edu.cn

**Abstract:** This paper constructs a two-sector manufacturer model of endogenous technological progress. We analyze the impact of environmental regulations on the factor input and output of different industries. Then, we reveal the intermediary role of inter-industry factor allocation in the impact of environmental regulations on industrial green total factor productivity (GTFP). Finally, the paper uses panel data from 30 provinces in China's industry from 2000 to 2017 to conduct empirical tests. We can draw the following conclusions: (1) The relative magnitude of the output compensation of the production department and the innovation compensation of the R&D department could change the impact of environmental regulations on the input and output of inter-industry factors, and the comprehensive effects of both input and output will affect the level of GTFP. (2) The curve of the direct impact of environmental regulations on GTFP is in an inverted "U" shape. However, the production factor allocation ratio can "reverse" the inhibitory effect of high-intensity regulations on GTFP. (3) The capital factor has a greater impact on the regulatory effect, but the labor factor has a more lasting impact on the regulatory effect. High-strength environmental regulations can enhance manufacturers' preference for human capital. Therefore, formulating environmental regulatory policies oriented to improve the ratio of factor allocation, mixing different types of regulatory policies, and increasing investment in human capital are all conducive to accelerating the transformation and upgrading of China's industrial structure and achieving high-quality development of the industrial economy.

**Keywords:** environmental regulation; green total factor productivity; inter-industry factor allocation; EBM-ML model; instrumental variable method

## 1. Introduction

Since the reform and opening up, China's economy has achieved rapid development. From 1978 to 2017, the average annual growth rate of China's gross domestic product (GDP) was 9.61%. In 2016, China developed into the second largest economy in the world, and the economic gap with the United States was narrowing. Although the rapid economic growth for more than 40 years has caused China's GDP to grow rapidly, it has also brought about serious environmental problems and hindered further economic growth. The report of the 19th National Congress of the Communist Party of China proposed that we should focus on high-quality economic development. Therefore, how to coordinate the relationship between economy and ecology and improve the quality of economic growth, while ensuring economic growth, is the primary task facing China's development at this stage. The essence of improving the quality of economic growth is to improve energy use efficiency and reduce environmental pollution during economic development, so as to realize the coexistence of gold and silver mountains and green mountains, that is, to improve green total factor productivity (GTFP). In order to control environmental pollution,

governments have promulgated and implemented various environmental regulations to reduce environmental pollution to a certain extent. However, some environmental governance measures are at the expense of development or the transfer of high-polluting industries, which deviates from the original intention of improving the quality of economic growth. Therefore, it is worth exploring the use of environmental regulations to reduce pollution while increasing GTFP.

The impact of environmental regulations on GTFP has always been the focus of the academic community. At present, there are mainly the following three views. The first view is that environmental regulations will increase the cost of environmental compliance for companies, reduce energy efficiency and corporate performance, and reduce GTFP, which is called the "cost effect" of environmental regulations [1]. The second view is that reasonable environmental regulations will force innovation by offsetting the increased costs of regulations to promote technological progress, improve energy efficiency, and improve GTFP, which is called the "compensation effect" of environmental regulations. [2–5]. The third view is that cost effect and compensation effect exist at the same time, and the impact of environmental regulation on GTFP depends on which effect is dominant under different intensity environmental regulations. On the one hand, the "cost effect" in the short term will crowd out the investment in innovation and reduce the productivity of enterprises. In the long run, rational manufacturers will use the "compensation effect" of technological innovation to offset the increase in costs caused by the "cost effect" and increase the productivity of enterprises, such as improving production and pollution control technologies. Therefore, a certain intensity of environmental regulations can stimulate technological innovation and improve GTFP [6,7]. Environmental regulations and GTFP have a "U"-shaped nonlinear relationship. On the other hand, high-strength environmental regulations will affect production activities. There is an inverted "U"-shaped change trend between environmental regulations and GTFP [8]. The authors in [9] concluded that environmental regulations should be within a suitable range, and that too high or too low levels of regulation are not conducive to the improvement of GTFP. In summary, the existing literature mainly focuses on the factor input and changes in input technology and research and development (R&D) factors caused by environmental regulations. However, GTFP is an efficiency indicator, which is affected by both factor input and output. Only by considering the changes in input and output can the impact of environmental regulations on GTFP be more accurately described.

Increasing factor input can promote economic growth [10]. With the advancement of supply-side reforms, the contribution rate of factor input to economic growth has maintained a downward trend for a long period of time [11]. The core driving force of economic growth needs to transform to increase productivity gradually. Studies have shown that the increase in productivity of an economy comes from the increase in productivity in various sectors of the economy and the increase in allocation efficiency caused by the flow of production factors among various sectors. The "mushroom effect" of the allocation of production factors from low-efficiency industries to high-efficiency industries can promote economic growth [12], because the factor replacement effect can reduce the misallocation of resources and improve the average quality of input factors [13]. Therefore, structural adjustment between industries is as important as technological innovation within the industry. Promoting the effective allocation of production factors, such as capital and labor among industries, is an effective means to improve the efficiency of the entire economic allocation [14]. As a type of industrial policy, the purpose of environmental regulations is to improve the environmental quality while not reducing the level of economic development. Its essence is to promote the transformation of industrial structure through technological progress. Therefore, in the process of environmental regulations to promote industrial GTFP, the role of factor allocation cannot be ignored. The existing research on the impact of environmental regulations on industrial GTFP mostly focuses on the impact on technological innovation, ignoring that the preference of innovative activities for elements will promote the allocation of capital and labor among different industries. Due to the difference

in pollution levels between pollution-intensive industries and clean industries, the actual effects of environmental regulations are also different. When the intensity of environmental regulations changes, factor input to different types of industries will change the factor allocation ratio because of the "prosperity to avoid disadvantages", which will change the GTFP of each industry and gradually change the GTFP level of the entire industry. Then, changes in the ratio of factor allocation between industries will also change the size of the impact of environmental regulations on industrial GTFP. The conclusion that the effect of environmental regulation on industrial GTFP is mainly achieved through technological innovation holds for the industry as a whole. However, due to the differences in the level of emissions in each industry, there are also differences in the level of technology used to combat pollution, which leads to the fact that the effect of environmental regulation on industrial GTFP through influencing technological innovation differs between clean and pollution-intensive industries. Technological innovation is affected by factor inputs, leading to an increase in the level of technology with the increase of factor inputs within a certain range. It is assumed that the total factor inputs remain constant over a certain period of time. Therefore, the change in the proportion of factor inputs between industries is equivalent to the change in the amount of factor inputs in different industries. When the ratio of factor inputs among different industries differs, the level of technological innovation in the two types of industries changes, thus changing the impact of environmental regulation on industrial GTFP. This paper focus on the factor allocation between industries and comprehensively studies the GTFP effect of environmental regulation from the two aspects of input and output to the industry level. This research helps to intuitively reflect the impact of environmental policies on the transformation of industrial structure and provides references for the formulation of government environmental policies. The research method diagram is shown in Figure 1.

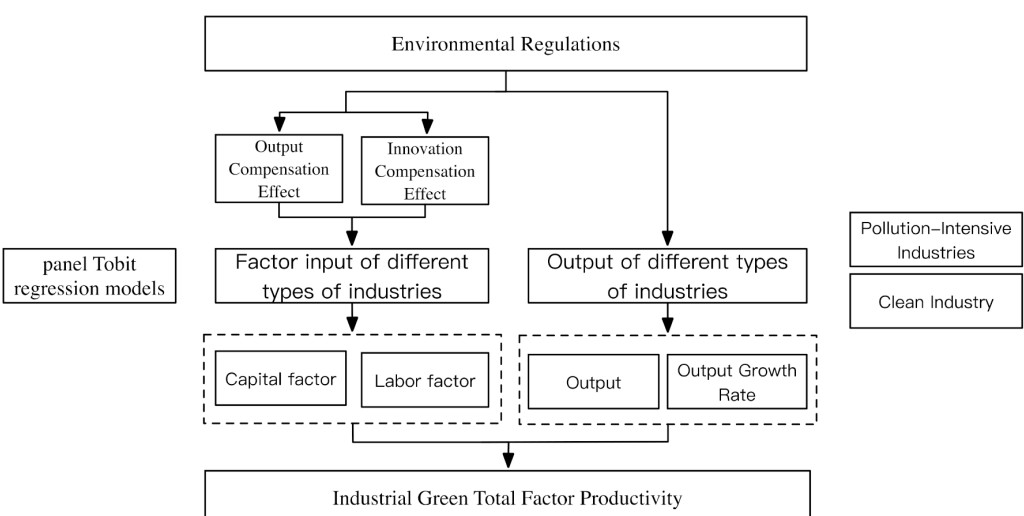

**Figure 1.** The research method diagram.

This study takes energy conservation, emission reduction, and economic development mode transformation as the entry points and provides an in-depth analysis of the impact of environmental policies on the production decision behavior of enterprises from the perspective of factor allocation among industries. By constructing a vendor model of endogenous technological progress, the paper reveals the micro mechanism of environmental regulation affecting industrial green total factor productivity and expands and improves the theory that environmental policy promotes economic growth quality improvement and economic transformation. At the same time, this study takes enterprises as decision makers and studies the behavioral choices of enterprises under government intervention from the perspective of factor allocation among industries, which points out the direction for industrial enterprises in economic transition. The findings of this study are beneficial for policy

makers to fully understand the role pathways in the process of policy implementation and provide references for future environmental policy formulation.

The contributions of this article mainly include the following three points. First, comparing factor input and industry output between industries, it is found that there is a difference between the direct impact and the indirect impact by affecting the ratio of factor allocation among different industries of environmental regulations on GTFP. The improvement of the distribution ratio of factors in different industries can "reverse" the inhibitory effect of high-strength environmental regulations on industrial GTFP. Second, this paper makes technological progress endogenous to the production function of the enterprise and establishes a two-sector model of the enterprise. It is found that the relative magnitude of the "output compensation effect" and "innovation compensation effect" affects the effect of environmental regulation on industrial GTFP. This study reveals the micro-mechanism of the impact of environmental regulations on industrial GTFP. The existing studies generally believe that environmental regulations can squeeze out the cost of innovation in a short period of time, ignoring the behavior of enterprises as an "economic man" who will try to reduce the cost of regulations. This article expands related research and improves the explanatory power of existing theories. Third, this paper uses Python to map wind speed and atmospheric altitude data to 30 provinces in China with the help of meteorological latitude and longitude data released by the European Center for Medium-Term Weather Forecast (ECMWF) and constructs the air flow coefficient as an instrumental variable for environmental regulations. It alleviates the endogenous problems in existing studies.

## 2. Theoretical Model

### 2.1. Benchmark Model

This paper establishes a two-sector manufacturer model to analyze the impact of environmental regulations on GTFP by affecting the allocation ratio of influencing factors among different industries. The premises of the research are as follows. First, the initial capital and labor input ratios of manufacturers belonging to different sub-sectors in the same industry are different, and the characteristics of different companies in the same sub-sector are the same. Second, each manufacturer has an R&D sector and a production sector. There is skill improved and the marginal product of factors is increased in the R&D sector. The technology produced by the R&D department continues to be invested in the production sector as an intermediate product. The marginal output of factors in the production sector is diminishing. Third, manufacturers will take the lead in changing labor input, because labor is more mobile than capital. Fourth, both product and factor markets are perfectly competitive, and manufacturers are price takers in the market.

According to the level of pollution emissions, this paper divides the sub-industries of industry into pollution-intensive industries and clean industries. Both types of industries have production sectors and R&D sectors. It is necessary to set the form of the production function of the two sectors. The Cobb–Douglas production function and the CES production function are more commonly used. Within the two sectors, the input factors are substitutable for each other, and in fact, the CES function is more realistic. However, the elasticity of substitution in the CES function is an uncertain parameter, which will increase the complexity of the analysis. Therefore, to simplify the analysis, this paper uses the C–D production function to set the elasticity of substitution to one.

The R&D sector uses capital $K_{12}$ and labor $L_{12}$ as inputs, and it uses technology as the output. There is technological progress. Pollution control technology is a function of environmental regulation $A_1(R)$. The production function of the R&D sector of a manufacturer in a pollution-intensive industry is as follows:

$$T_1 = A_1(R)K_{12}{}^{\alpha_{12}}L_{12}{}^{\beta_{12}}, \tag{1}$$

where $\alpha_{12}$ and $\beta_{12}$ are the output elasticity of the invested capital and labor of the R&D sector. The production sector invests capital $K_{11}$, labor $L_{11}$, technology $T_1$, and energy

resources $E_1$ to produce the final product, and technology is the output of the R&D sector. The production sector will discharge pollution $W_1$ during production, which will affect production activities as a negative output. In this paper, pollution is introduced into the production function as a negative impact on output, expressed as $[1 - d(W_1)]$, referring to the research of Fan Qingquan [15] and Tong Jian et al. [16].

The production function of the production sector of a manufacturer in a pollution-intensive industry is as follows:

$$Y_1 = [1 - d(W_1)]T_1 K_{11}{}^{\alpha_{11}} L_{11}{}^{\beta_{11}} E_1{}^{\gamma_1}, \tag{2}$$

$$W_1 = W_1(E_1, R, A_1(R)) = \frac{\rho_1 E_1}{A_1(R)R} \tag{3}$$

where $\alpha_{11}$, $\beta_{11}$, and $\gamma_1$ are output elasticity of capital, labor, and energy, respectively. $Y_1$ and $R$ are output and environmental regulations, respectively. $d(W_1)$ represents the negative impact of pollution on output and increases with the increase of pollution. $W_1$ represents the amount of pollution emissions, which is inversely proportional to the level of pollution control technology and directly proportional to the amount of energy input, and $\rho_1$ is the proportional coefficient.

Similar to pollution-intensive industries, the production functions of manufacturers' R&D sectors and production sectors of clean industries are as follows:

$$T_2 = A_2(R)K_{22}{}^{\alpha_{22}} L_{22}{}^{\beta_{22}}, \tag{4}$$

$$Y_2 = [1 - d(W_2)]T_2 K_{21}{}^{\alpha_{21}} L_{21}{}^{\beta_{21}} E_2{}^{\gamma_2} \tag{5}$$

$$W_2 = W_2(E_2, R, A_2(R)) = \frac{\rho_2 E_2}{A_2(R)R} \tag{6}$$

where the interpretation of variables is similar to that of pollution-intensive industries. Both pollution-intensive industries and clean industries have positive technical levels, but clean industries discharge less pollution due to higher levels of pollution control technology and the productivity of R&D sectors. It is constrained to $0 < A_1(R) < A_2(R)$, $T_1 < T_2, W_1 > W_2$. With the improvement of the intensity of environmental regulations, manufacturers in the clean industry will continuously adjust their pollution control technology level according to the changes in the level of regulation, so as to maintain the emission of pollution at a low level. The pollution control technology of manufacturers in pollution-intensive industries will not change with changes in environmental regulations. It will maintain the original technical level for production. The pollution control technology level of the two industries is shown as $0 = A'_1(R) < A'_2(R)$. When pollution emissions increase, clean industries can treat more pollution and have less negative impact on output due to their higher level of pollution control technology. Pollution-intensive industries have a lower level of pollution control to deal with less pollution, so they have a greater negative impact on output. It is shown as $d(W_1) > d(W_2)$, and $d'_1(W_1) > 0, d'_2(W_2) > 0$.

*2.2. The Impact of Environmental Regulations on the Allocation Ratio of Factors among Industries*

According to the firm theory, analyzing the choice of the firm requires the first-order conditions for maximizing profit. The profit of a manufacturer is shown in the following equation:

$$\Pi = PY_1 - C(K, L, R), \tag{7}$$

where $\Pi$ represents the profit of a manufacturer and $K$ and $L$ are all capital and labor invested by the manufacturer, respectively. $P$ represents product price. The cost function $C(K, L, R)$ is constrained to $C'_{K_1}(K, L, R) > 0$ and $C'_{K_2}(K, L, R) > 0$, where $K_1$ and $K_2$ represent total capital investment in pollution-intensive and clean industries, respectively. A rational manufacturer will use input elements to maximize its profits. Use $K_1$ and $K_2$ to find the partial derivatives of Equation (7), and substitute Equations (1) and (2) into it.

The first-order conditions for maximizing profit are shown in the following equations:

$$\frac{\partial \Pi}{\partial K_1} = P[1 - d(W)]A(R)K_2{}^{\alpha_2}L_2{}^{\beta_2}\alpha_1 K_1{}^{\alpha_1-1}L_1{}^{\beta_1}E^{\gamma} - C'_{K_1}(K, L, R) = 0, \quad (8)$$

$$\frac{\partial \Pi}{\partial K_2} = P[1 - d(W)]A(R)\alpha_2 K_2{}^{\alpha_2-1}L_2{}^{\beta_2}K_1{}^{\alpha_1}L_1{}^{\beta_1}E^{\gamma} - C'_{K_2}(K, L, R) = 0 \quad (9)$$

Divide Equation (9) by Equation (8) to obtain the following equation:

$$\frac{\alpha_2 K_1}{\alpha_1 K_2} = \frac{C'_{K_2}(K, L, R)}{C'_{K_1}(K, L, R)}, \quad (10)$$

By incorporating the relevant variables of pollution-intensive industries and clean industries into Equation (10), we can obtain the following equations for pollution-intensive industries and clean industries:

$$\frac{\alpha_{12}K_{11}}{\alpha_{11}K_{12}} = \frac{C'_{K_{12}}(K_1, L_1, R)}{C'_{K_{11}}(K_1, L_1, R)}, \quad (11)$$

$$\frac{\alpha_{22}K_{21}}{\alpha_{21}K_{22}} = \frac{C'_{K_{22}}(K_2, L_2, R)}{C'_{K_{21}}(K_2, L_2, R)}, \quad (12)$$

Assume that the marginal product of capital and labor of the two types of industries are the same, respectively, and satisfy $K_1 = K_{11} + K_{12}$, $L_1 = L_{11} + L_{12}$, $K_2 = K_{21} + K_{22}$, $L_2 = L_{21} + L_{22}$. Divide Equation (11) by Equation (12) and substitute $K_{11}$ and $K_{21}$ into it, and the equation is as follows:

$$\frac{K_{11}(K_2 - K_{21})}{K_{21}(K_1 - K_{11})} = \frac{C'_{K_{12}} \cdot C'_{K_{11}}}{C'_{K_{22}} \cdot C'_{K_{21}}}, \quad (13)$$

Decompose the left side of Equation (13) to obtain the following formula:

$$\frac{K_{11}(K_2 - K_{21})}{K_{21}(K_1 - K_{11})} = \frac{K_{11} \cdot K_2 - K_{11} \cdot K_{21}}{K_{21} \cdot K_1 - K_{21} \cdot K_{11}} > 1 - \frac{K_{11} \cdot K_2}{K_{21} \cdot K_1}, \quad (14)$$

Substituting Equation (13) into formula (14) gives the following formula:

$$\frac{K_1}{K_2} < \frac{K_{11}}{K_{21}}\left[1 - \frac{C'_{K_{22}} \cdot C'_{K_{21}}}{C'_{K_{12}} \cdot C'_{K_{11}}}\right], \quad (15)$$

Set $V = \frac{K_1}{K_2}$ to obtain the following formula:

$$\frac{\partial V}{\partial R} < \frac{K_{11}}{K_{21}}\left[1 - \frac{C''_{K_{12},R} \cdot C'_{K_{21}} + C''_{K_{21},R} \cdot C'_{K_{12}}}{C'_{K_{12}} \cdot C'_{K_{21}}} - \frac{C''_{K_{22},R} \cdot C'_{K_{11}} + C''_{K_{11},R} \cdot C'_{K_{22}}}{C'_{K_{22}} \cdot C'_{K_{11}}}\right] < \frac{K_{11}}{K_{21}}\left[1 - \left(\frac{C''_{K_{12},R}}{C'_{K_{12}}} + \frac{C''_{K_{11},R}}{C'_{K_{11}}} + \frac{C''_{K_{21},R}}{C'_{K_{21}}} + \frac{C''_{K_{22},R}}{C'_{K_{22}}}\right)\right] \quad (16)$$

where the items in the parentheses on the right represent the ratio of the marginal cost of capital of the production department and the R&D department, which invest in the production sectors and R&D sectors of pollution-intensive industries and clean industries when regulations are strengthened and unchanged, respectively. When environmental regulations are strengthened, the marginal cost of factors increases because manufacturers will spend part of the money to control pollution. Therefore, the numerators in the parentheses on the right side of formula (16) are greater than the denominator, resulting in $\partial V/\partial R < 0$. That is, stronger environmental regulations enable more production factors to be allocated to clean industries. This leads to Hypothesis 1.

**Hypothesis 1.** *With the enhancement of environmental regulations, the allocation of capital elements to clean industries has led to a decline in the proportion of capital allocation in pollution-intensive industries and clean industries.*

### 2.3. The Impact of Environmental Regulations on the Output of Different Types of Industries

Using $R$ to find the partial derivatives of Equation (2), the impact of environmental regulations on output is shown in the following equation:

$$\frac{\partial Y_1}{\partial R} = \left\{ d'(W_1)\rho_1 E_1 \cdot \left[ \frac{A'_1(R)}{A_1(R)R} + \frac{1}{R^2} \right] + A'_1(R)[1 - d(W_1)] \right\} K_{11}{}^{\alpha_{11}} L_{11}{}^{\beta_{11}} K_{12}{}^{\alpha_{12}} L_{12}{}^{\beta_{12}} E_1{}^{\gamma_1} \tag{17}$$

Since $A'_1(R) = 0$, we obtain the following formula:

$$\frac{\partial Y_1}{\partial R} = \frac{d'(W_1)\rho_1 E_1}{R^2} K_{11}{}^{\alpha_{11}} L_{11}{}^{\beta_{11}} K_{22}{}^{\alpha_{22}} L_{22}{}^{\beta_{22}} E_1{}^{\gamma_1} > 0, \tag{18}$$

Using $R$ to find the partial derivatives of $\partial Y_1/\partial R$, the impact of the growth rate of environmental regulations on output is shown as follows:

$$\frac{\partial^2 Y_1}{\partial R^2} = -\frac{\rho_1 E_1}{R^3} d'(W_1) K_{11}{}^{\alpha_{11}} L_{11}{}^{\beta_{11}} K_{22}{}^{\alpha_{22}} L_{22}{}^{\beta_{22}} E_1{}^{\gamma_1} < 0, \tag{19}$$

Equations (18) and (19) show that, as the intensity of environmental regulations increases, the output of pollution-intensive industries gradually increases, but the output growth rate gradually decreases until it drops to zero. Therefore, when the intensity of environmental policy and regulation reaches a certain level, the output of pollution-intensive industries increases to the maximum. Figure 2 shows the relationship between the intensity of environmental regulations and the output of pollution-intensive industries.

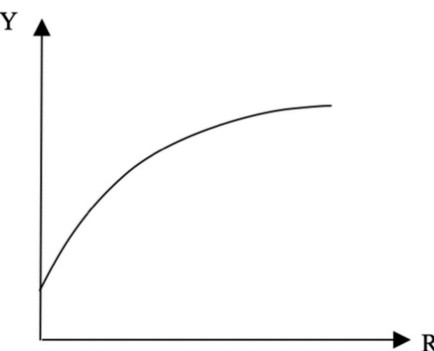

**Figure 2.** The relationship between the intensity of environmental regulations and output in pollution-intensive industries.

Similarly, the use of $R$ to find first-order and second-order partial derivatives on the output of clean industries will result in the following formulas:

$$\frac{\partial Y_2}{\partial R} = \left\{ d'(W_2)\rho_2 E_2 \cdot \left[ \frac{A'_2(R)}{A_2(R)R} + \frac{1}{R^2} \right] + A'_2(R)[1 - d(W_2)] \right\} K_{21}{}^{\alpha_{21}} L_{21}{}^{\beta_{21}} K_{22}{}^{\alpha_{22}} L_{22}{}^{\beta_{22}} E_2{}^{\gamma_2} > 0 \tag{20}$$

$$\begin{aligned} \frac{\partial^2 Y_2}{\partial R^2} = &\left\{ \rho_2 E_2 d'(W_2) \left[ \frac{A''_2(R)A_2(R)R - A'^2_2(R)R - A'_2(R)A_2(R)}{A^2_2(R)R^2} - \frac{2}{R^2} \right] + A''_2(R)[1 - d(W_1)] \right\} \\ &\cdot K_{21}{}^{\alpha_{21}} L_{21}{}^{\beta_{21}} K_{22}{}^{\alpha_{22}} L_{22}{}^{\beta_{22}} E_2{}^{\gamma_2} \end{aligned} \tag{21}$$

When the intensity of regulation is weak, the pollution discharge pressure of clean industries is smaller, so the driving force for technological progress is also weaker, and the growth of $A'_2(R)$ is slower. With the gradual increase in the intensity of regulations, the pressure on the clean industry to discharge pollution also increases. At this time, the driving

force for technological progress and the growth rate of $A'_2(R)$ are also gradually increasing, showing as $A''_2(R) > 0$. It can be converted to $A'_2(R) = A''_2(R) \cdot dR < A''_2(R) \cdot R$. As the same, $A_2(R) < A'_2(R) \cdot R$. The right side of Equation (21) can be rewritten as the Equation (23):

$$\frac{A''_2(R)A_2(R)R - A'^2_2(R)R - A'_2(R)A_2(R)}{A^2_2(R)R^2} - \frac{2}{R^2} > -3\left[\frac{A'_2(R)}{A_2(R)R}\right]^2 > -3\left[\frac{A''_2(R)}{A_2(R)}\right]^2, \tag{22}$$

$$\frac{\partial^2 Y_2}{\partial R^2} > A''_2(R)\left\{[1 - d(W_1)] - 3\rho_2 E_2 d'(W_2)\left[\frac{A''_2(R)}{A_2(R)}\right]\right\} \tag{23}$$

In the second term of Equation (23), $A_2(R)$ increases with the increase of $R$, and $A''_2(R)$ and $d'(W_2)$ are constants, so the term decreases as R increases. As $R$ increases further, the second term will gradually decrease to less than $[1 - d(W_1)]$, so that $\partial^2 Y_2/\partial R^2$ will change from negative to positive. Therefore, when the intensity of regulation gradually increases, the output of clean industries will gradually increase, but its growth rate will first decline and then rise. Figure 3 shows the relationship between the intensity of environmental regulations and the output of clean industries.

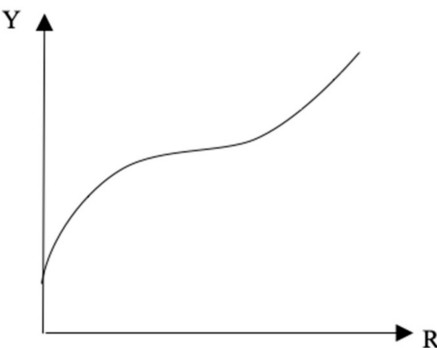

**Figure 3.** The relationship between the intensity of environmental regulations and output in clean industries.

There are both output compensation effects and innovation compensation effects in pollution-intensive industries and clean industries. The two effects make the output of the two types of industries gradually increase with the increase in the intensity of environmental regulations, but there are differences in the relative magnitude of the two effects in the two types of industries. Due to the low level of pollution control technology and innovation compensation in pollution-intensive industries, pollution control technology will remain unchanged at the original level when regulations are strengthened. The lower technical level is difficult to compensate for the increase in regulatory costs. Manufacturers will increase output by increasing the factor input of the production department, which is called an output compensation effect. Therefore, the output compensation effect of pollution-intensive industries is greater than the innovation compensation effect. The level of pollution control technology and innovation compensation for clean industries are relatively high. When regulations are strengthened, pollution control technologies will be upgraded to maintain a low level of pollution, which will bring about technological progress in the industry. Therefore, the innovation compensation effect of clean industries is greater than the output compensation effect.

When the output compensation effect is dominant, the cost of pollution discharge increases with the enhancement of environmental regulations, and it is becoming more and more difficult to use the increase in output to compensate for the cost of compliance. The output growth rate of pollution-intensive industries is getting slower and slower, and the output gradually increases to a certain level.

The compensation effect of innovation increases output by promoting technological progress. With the improvement of the intensity of regulation, the innovation compensation

effect is in a dominant position, which has continuously improved the level of pollution control technology. Because the output growth brought about by the improvement of the technological level is persistent, the output of the clean industry will continue to increase. This leads to Hypotheses 2 and 3.

**Hypothesis 2.** *With the enhancement of environmental regulations, the output of pollution-intensive industries has gradually increased, but its growth rate has decreased.*

**Hypothesis 3.** *With the enhancement of environmental regulations, the output of clean industries has gradually increased, and its growth rate first declined and then increased.*

*2.4. The Impact of Environmental Regulations on Industrial Green Productivity*

Comparing the impact of factor allocation and output of different industries on GTFP in different industries, the change trend of industrial GTFP can be obtained by adding the GTFP of the two types of industries. Hypothesis 1 shows that with the enhancement of environmental regulations, the capital investment in pollution-intensive industries and the capital investment in clean industries have increased, but the increase in capital investment in clean industries has been even greater. Assume that the increment of capital investment among different industries all grow linearly.

For the pollution-intensive industries, we focus on the extreme situation of Hypothesis 2: as the intensity of environmental regulations increases, the capital input of the industry increases linearly while the increment of output decreases to zero. It indicates that the input–output efficiency of pollution-intensive industries, namely GTFP, decreases with the increase of capital input. Therefore, the GTFP of pollution-intensive industries gradually declines with the increase of environmental regulation intensity. For clean industries, it is known from Hypothesis 3 that their output shows a non-linear growth trend. The level of environmental regulation at the output growth rate of zero is taken as the cut-off point. On the left side of the cut-off point, similar to the pollution-intensive industry, the capital input of the industry increases linearly with the increase of environmental regulation intensity, while the increment of output decreases gradually until it reaches zero. The input–output efficiency of the clean industry, namely GTFP, gradually decreases with the increase of capital input. On the right side of the cut-off point, with the increase of the intensity of environmental regulation, the capital input of the industry still increases linearly, but the output increases non-linearly. The input–output efficiency of the clean industry, namely GTFP, gradually increases with the increase of capital input. Thus, the GTFP of clean industry decreases and then increases with the increase of environmental regulation intensity, showing a "U"-shaped trend. Considering the trends of GTFP in pollution-intensive industries and clean industries, the industrial GTFP shows a "U"-shaped trend of decreasing and then increasing in general. This leads to Hypothesis 4.

**Hypothesis 4.** *With the enhancement of environmental regulations, the allocation of capital between pollution-intensive industries and clean industries has declined, leading to a "U"-shaped trend in industrial GTFP.*

## 3. Model and Variables

*3.1. Regression Model*

Equations (24) and (25) are ordinary panel regression models to verify the relationship between environmental regulations and industry output as well as output growth rate.

$$lnY_{it} = \theta_0 + \theta_1 Enr_{it} + \theta_2 Z_{it} + \varphi_{it}, \tag{24}$$

$$ln\dot{Y}_{it} = \theta'_0 + \theta'_1 Enr_{it} + \theta'_2 Enr^2_{it} + \theta'_3 Z_{it} + \varphi_{it} \tag{25}$$

where $lnY_{it}$ and $ln\dot{Y}_{it}$ denote the industry output and output growth rate in year $t$ of region $i$, respectively; $Enr_{it}$ denotes the level of environmental regulation in year $t$ of region $i$, and $Z_{it}$ is all control variables. The maximum likelihood estimate (MLE) was used to estimate the model to ensure the consistency of the estimation results. Equations (24) and (25) are used to test Hypotheses 1 and 2, respectively.

A three-stage model is used to test the indirect effect of environmental regulation on GTFP by changing the ratio of factor allocation among different industries. Considering the explanatory variable industrial GTFP as a restricted variable, the last two models were replaced with panel Tobit regression models.

$$K\_outflow_{it} = \alpha_0 + \alpha_1 Enr_{it} + \alpha_2 Z_{it} + \varepsilon'_{it}, \tag{26}$$

$$GTFP^*_{it} = \beta_0 + \beta_1 Enr^2_{it} + \beta_2 Enr_{it} + \beta_3 Z_{it} + \vartheta'_{it} \tag{27}$$

$$GTFP^*_{it} = \gamma_0 + \gamma_1 Enr^2_{it} + \gamma_2 Enr_{it} + \gamma_3 K\_outflow_{it} + \gamma_4 K\_outflow_{it}^2 + \gamma_5 Z_{it} + \varphi'_{it} \tag{28}$$

$$GTFP_{it} = \begin{cases} 1 & if\ GTFP^*_{it} > 1 \\ GTFP^*_{it} & if\ 0 < GTFP^*_{it} \leq 1 \\ 0 & if\ GTFP^*_{it} \leq 0 \end{cases} \tag{29}$$

where $K\_outflow_{it}$ denotes the inter-industry capital factor allocation ratio in year $t$ of region $i$ ($K\_outflow_{it}$ is replaced by $L\_outflow_{it}$ to study the inter-industry labor factor allocation ratio), $GTFP_{it}$ denotes the actual industrial GTFP in year $t$ of region $i$, $GTFP^*_{it}$ is the latent variable, and other variables have the same meaning as above. Considering the consistency of the estimation results, the regression model is estimated by the method of MLE. Equation (26) is used to test Hypothesis 3, where the coefficient $\alpha\_1$ measures the effect of environmental regulation on the factor allocation ratio. Equation (27) verifies the overall effect of environmental regulation on industrial GTFP. The coefficients $\gamma_1$ and $\gamma_2$ of Equation (28) measure the direct impact of environmental regulation on industrial GTFP. Equations (26) and (28) are used to test Hypothesis 4, whose coefficients $\alpha_1$, $\gamma_3$, and $\gamma_4$ together measure the indirect effect of environmental regulation on industrial GTFP by affecting the factor allocation ratio.

### 3.2. Measurement of GTFP

In this paper, we adopt a non-oriented EBM-ML model with non-consensual outputs to measure industrial GTFP, which can maximize profits by considering both input reduction and output increase and can effectively avoid the productivity overestimation of the oriented model. The industrial GTFP measured in this paper uses capital, labor, and energy as inputs and produces finished products while discharging pollution. Each input–output indicator is selected as follows.

### 3.2.1. Capital

Most of the existing studies set the depreciation rate to a constant value when calculating capital investment. However, depreciation rates can vary depending on the equipment purchased, the region in which it is located, and the particular year. Therefore, this paper returns to the concept of the basic perpetual inventory method and calculates the depreciation rate using the ratio of the current year's depreciation to the previous year's original cost of fixed assets. Moreover, the initial capital stock will have an impact on the capital stock in subsequent years. In this paper, drawing on Tu Zhengge [17] and Pang Ruizhi and Li, Peng [18], the capital stock is approximated using the annual average balance of the net fixed asset investment of industrial enterprises above the size of each province, and the net fixed asset investment calculated in previous years is adjusted to comparable price data, with the year 2000 as the base period using the fixed asset investment index. The data are obtained from the *China Industrial Economic Statistical Yearbook*.

### 3.2.2. Labor

The number of employees at the end of the year in industrial enterprises above the size of each province is used to measure labor. The data were obtained from the *China Industrial Economy Statistical Yearbook*, where the year-end number of employees in 2004 was obtained from the 2004 *China Economic Census Yearbook*.

### 3.2.3. Energy

The energy consumption of industrial enterprises above the size of each province was used to measure energy. The data are obtained from the energy consumption of each region in the *China Energy Statistical Yearbook*. In this paper, the unit of energy consumption is converted to million tons of standard coal using the conversion coefficient, which is obtained from the *China Energy Statistical Yearbook* of previous years.

### 3.2.4. Expected Output

The expected output is generally measured using industrial GDP [3,19] or industrial value added [20,21] in existing studies. Since the energy in the input indexes has the nature of intermediate inputs when calculating industrial GTFP in this paper, it is appropriate to use the industrial GDP that includes the cost of intermediate inputs for the expected output [19]. At the same time, considering the availability of data and the consistency across years, the total industrial output value of each province is used as the expected output indicator in this paper. In order to eliminate the price factor, the industrial GDP of each province is deflated by using the ex-factory industrial producer price index of each province separately, using the year 2000 as the base period. The data are obtained from the *China Industrial Economic Statistical Yearbook*.

### 3.2.5. Non-Expected Output

The EBM-ML is a relative efficiency accounting method. This means that when the measurement indicators of each variable remain relatively consistent, the results will not have large deviations [22]. Considering the availability of data, in this paper, the industrial wastewater emissions, industrial sulfur dioxide emissions, and industrial smoke (dust) emissions of each province are combined into one pollution index to measure the industrial non-desired output of each province. As for the methods of comprehensive evaluation, there are principal component analysis, factor analysis, expert scoring method, and entropy value method. The entropy value method determines the indicator weights based on the relative changes of the impact of the indicators on the overall system, which can reflect the effect and value of the indicators. It is similar to the effect of the role of each indicator in the pollution index, in which the main factors affecting the pollution index are also those with a large degree of variation. Therefore, this paper adopts the entropy value method to estimate the weights of each indicator in the pollution index and then the weighted average of three kinds of pollution emissions to obtain the comprehensive pollution index, which is used to measure the industrial pollution emission level of each province in China. The pollutant emission data are obtained from the *China Environmental Statistical Yearbook* and the *China Environmental Yearbook*. This paper uses MaxDEA 8 Ultra software to measure the industrial GTFP of 30 provinces of China in calendar years except Tibet.

### *3.3. Variable Selection*

### 3.3.1. Environmental Policies and Regulation

The intensity of environmental regulations represents the severity of environmental regulations in various regions. In empirical research, scholars at home and abroad mainly use six methods to measure the intensity of environmental regulations, such as the promulgation and implementation of environmental policies in various regions, the amount of pollution emitted per unit of production, the costs or expenses incurred during the operation of pollution control facilities in various regions, per capita pollution control expenses, the proportion of the company's total cost or total output value of the company's

pollution control costs, per capita income, and a comprehensive index constructed based on the emissions of three main pollutants (exhaust gas, wastewater, and solid waste). However, all these methods have certain shortcomings. The number of policies only reflects the importance of environmental protection at all levels of government, but not the level of implementation of the relevant policies. In addition, data on the implementation efforts are difficult to obtain. The amount of pollution emitted per unit of product is greatly influenced by product heterogeneity. The operating cost of pollution control in each region is influenced by the size of local enterprises and the industry they belong to, and the data are not comparable between regions. The proportion of pollution control costs of enterprises is affected by the heterogeneity of enterprises, for example, the proportion of pollution control costs of enterprises in low-pollution industries is lower, while the proportion of pollution control costs of enterprises in high-pollution industries tends to be higher. The relationship between per capita income and the intensity of environmental regulations is not simply linear; as per capita income increases, people become less tolerant of the environment. The composite index constructed using pollutant emissions only reflects the absolute quantity of pollution emissions, ignoring the heterogeneity of regions and industries.

Based on the above analysis, the ideal proxy variable for environmental regulation should both characterize the absolute level of pollution emissions of enterprises within the region and reflect the difference between the emissions of the region and other regions. Considering the data availability and data quality, this paper constructs the following composite indicators as the proxy variables of environmental regulation.

$$Enr_{it} = \frac{1}{3} \sum_{j=1}^{3} \left( v_{ij} / \frac{1}{30} \sum_{i=1}^{30} v_{ij} \right), \tag{30}$$

where $v_{ij}$ is the industrial value added for the region $i$ by emitting one unit of the pollutant $j$, which indicates the relative level of the output value per unit of pollution emission of a region in the country. The higher the industrial value added and the lower the pollution emission, the higher the intensity of environmental regulation in the region. Meanwhile, the unit pollution control investment is used as a robustness test. The unit pollution control investment is calculated by dividing the regional pollution control investment by the pollution composite index. The higher pollution control cost a region invests in pollution control, the higher the intensity of environmental regulation.

### 3.3.2. Output of Different Industries

The division of pollution-intensive industries and clean industries was carried out by drawing on the study of Ling Li and Feng Tao [23]. First, the pollution emissions per unit of output value of wastewater, exhaust gas, and solid waste were calculated for each segment of industry. Then, the pollution emission values per unit output value of the three pollutants for each industry were linearly normalized according to a range of values from 0 to 1. Finally, the above three pollution emission scores are arithmetically averaged to obtain the total pollution emission intensity factor $\gamma$ for the industry. The median of the pollution emission intensity coefficient ($\gamma_m$) is used as the standard. If $\gamma < \gamma_m$, then it is a clean industry; if $\gamma > \gamma_m$, then it is a pollution-intensive industry. With the development of the economy and the adjustment of industrial structure, the ranking of pollution emissions of various industries in different places has changed in the past years. Therefore, in this paper, the sub-sectors of pollution-intensive and clean industries are obtained for 30 provinces across the country from 2000 to 2017, respectively, by pollution emission intensity for different years. Therefore, this paper divides each industry by pollution emission intensity within different years to obtain the breakdown of pollution-intensive and clean industries in 30 provinces across China from 2000–2017. Limited to the availability of pollutant emissions by industry by province, provincial differences are not considered in this paper.

Referring to the above classification criteria, the sales value of each industrial sub-sector belonging to pollution-intensive industries is summed up as the output of pollution-intensive industries. The growth rate of output is calculated by using (current year's sales

output − previous year's sales output)/previous year's sales output. Similarly, the output and output growth rate of clean industries can be obtained.

### 3.3.3. Allocation Ratio of Elements among Industries

For the capital factor ($K\_outflow_{it}$), the fixed asset investments belonging to pollution-intensive and clean industries are summed up separately and converted to capital stock using the perpetual inventory method (the same method as before). Then, the capital stock $K_{1it}$ of pollution-intensive industries is divided by the capital stock $K_{2it}$ of clean industries as the allocation ratio of capital factors among industries. An increasing ratio indicates that capital is invested more in pollution-intensive industries, and vice versa indicates that it is invested more in clean industries.

For the capital element ($L\_outflow_{it}$), the year-end number of employees in the corresponding industries is summed up separately according to the industry classification. Then, the year-end number of employees $L_{1it}$ of pollution-intensive industries is divided by the year-end number of employees $L_{2it}$ of clean industries as the allocation ratio of labor factors among industries. An increasing ratio indicates that labor is invested more in pollution-intensive industries, and vice versa indicates that it is invested more in clean industries.

### 3.3.4. Control Variables

Referring to existing studies, this paper selects technological progress [24], property rights structure [25], industrial agglomeration [26], energy consumption structure, economic development level, wage level [27], and foreign direct investment [28] as the control variables of the model. The measurement of each variable is shown in Table 1.

**Table 1.** Description of control variables.

| Variable Name | Proxy Variables | Measurement Method |
|---|---|---|
| Technological Progress ($TEC_{it}$) | Effective invention patents for enterprises. | Number of valid invention patents for industrial enterprises in each region. |
| Property Rights Structure ($PRO_{it}$) | The proportion of output value of state-owned enterprises. | The output value of state-owned and state-controlled industrial enterprises divided by the output value of industrial enterprises above the scale. |
| Industrial Agglomeration ($LQ_{it}$) | Zone entropy index. | $LQ_{it} = \dfrac{x_{it}/\sum_i x_{it}}{\sum_n x_{it}/\sum_i \sum_n x_{it}}$ |
| Energy Consumption Structure ($EST_{it}$) | Percentage of coal energy consumption. | Coal consumption divided by total energy consumption. |
| Economic Development ($GDPP_{it}$) | GDP per capita. | Standardized regional GDP per capita. |
| Wage ($WAG_{it}$) | Average wage. | Standardized regional average wage of industrial enterprises. |
| Foreign Direct Investment ($FDI_{it}$) | Percentage of actual foreign investment utilized. | Actual utilization of foreign investment by industrial enterprises divided by industrial value added. |

### 3.4. Data Source and Processing

The research object of this paper is industry. Data from 2000 to 2017 are selected for 30 provinces across China, excluding Tibet, where there are serious data deficiencies. The time span is selected mainly for the following three reasons. First, the inflection point in the regional allocation of economic resources by the state in 2000, e.g., from 2000, the state allocated more construction land targets and transfer payments to the central and western regions [29]. Using only the data after this inflection point prevents policy shocks from biasing the estimation of the econometric model. Second, to ensure the completeness of all the data available in this paper, the cut-off year of the data is 2017, because the data related

to industrial wastewater, industrial sulfur dioxide, and industrial solid waste emissions are not available in the *China Environmental Statistics Yearbook 2019* and subsequent yearbooks. Third, the longer the time span the more likely to be disturbed by policy changes, and the greater the possibility of bias in the regression analysis, because China is in an era of change.

The data related to environmental regulation, inter-industry factor allocation ratio, and control variables are obtained from the *China Statistical Yearbook*, *China Industrial Economic Statistical Yearbook*, *China Energy Statistical Yearbook*, *China Science and Technology Statistical Yearbook*, *Labor Statistical Yearbook*, and *Fixed Asset Investment Yearbook* of each province from 2001 to 2018. For the data of industrial enterprises above the scale, the ratios of industrial enterprises above the scale to all industrial enterprises in the *China Economic Census Yearbook* in 2004, 2008, and 2013 were collected, and the data of other years were completed by using Python fitting. The resulting data are then used to adjust each indicator to all industrial enterprises. In order to eliminate the influence of prices, this paper takes the year 2000 as the base period and uses price indices to convert the relevant indicators into the actual amounts calculated according to constant prices. Total assets were adjusted using the perpetual inventory method by drawing on the practice of Jun Zhang et al. [30].

## 4. Results and Discussion

### 4.1. The Impact of Environmental Regulations on the Allocation Ratio of Factors among Industries

Model (4) and model (5) in Table 2 show the effect of environmental regulations on the proportion of factors allocated among industries. The results show that as the intensity of environmental regulation increases, capital and labor factors are allocated more toward clean industries. Hypothesis 1 is tested. Production and R&D sectors exist within both pollution-intensive and clean industries. When the intensity of regulation starts to increase, the output compensation is greater than the innovation compensation, causing more allocation of factors to the production sector. Pollution-intensive industries discharge relatively more pollution, and their output compensation is smaller than that of technology-intensive industries, which implies that the marginal cost of factors in pollution-intensive industries is higher when the amount of factor inputs is the same in both industries. Therefore, rational manufacturers will allocate more capital and labor to clean industries, causing the allocation ratio of factors between industries to decrease. Therefore, rational manufacturers will allocate more capital and labor to clean industries, causing the allocation ratio of factors between industries to decrease. When the intensity of regulation increases further, factors are allocated more to the R&D sector because the innovation compensation is greater than the output compensation. The higher level of pollution control technology is in the clean sector, and its innovation compensation is smaller than that in the pollution-intensive sector, indicating that the marginal cost of factors in the clean sector is lower when the amount of factor inputs is the same in both sectors. Therefore, rational manufacturers will still allocate more capital and labor to clean industries, causing a further decrease in the factor allocation ratio between industries.

### 4.2. The Impact of Environmental Regulations on the Output of Different Types of Industries

A time-fixed effects panel model was used for estimation. The results of model (1) and model (3) in Table 3 show that as the intensity of environmental regulation increases, the output of pollution-intensive industries increases significantly, but the output growth rate decreases significantly. Hypothesis 2 is tested. The results of model (2) and model (4) show that as the intensity of environmental regulation increases, the output of clean industries increases significantly, and the output growth rate decreases and then increases in a "U"-shaped trend. Hypothesis 3 is tested. Since the output compensation effect of pollution-intensive industries is greater than the innovation compensation effect, the increase in regulation raises the cost of pollution emissions and makes it more and more difficult to compensate for the cost by increasing output. Therefore, the growth rate of output in pollution-intensive industries gradually decreases, and the output tends to

a specific level in the process of gradual increase. The innovation compensation effect of clean industries is greater than the output compensation effect, and the innovation compensation effect enhances output through technological progress. With the increase of regulation intensity, the innovation compensation effect is in the dominant position to make the technology level of pollution control continuously improve. The output of clean industry will continue to increase because of the durability of the output growth brought by technological advancement. However, the initial stage of technological innovation requires a large amount of investment, which squeezes out some production inputs in the short term, thus causing a decrease in the output growth rate. At a later stage, production inputs are supplemented while innovations bring about technological advances, causing output growth rates to increase.

**Table 2.** Description of control variables.

| Variables | GTFP | | | $K\_outflow_{it}$ | $L\_outflow_{it}$ |
|---|---|---|---|---|---|
| | **(1)** | **(2)** | **(3)** | **(4)** | **(5)** |
| $ENR_{it}$ | 0.1999 ***(3.06) | 0.1825 ***(2.80) | 0.1987 ***(3.06) | −0.0013 *(−1.67) | −0.0299 *(−1.81) |
| $ENR_{it}^2$ | −0.1325 **(−2.07) | −0.1164 *(−1.82) | −0.1300 **(−2.04) | - | - |
| $K\_outflow_{it}$ | - | 0.0283 **(2.05) | - | - | - |
| $K\_outflow_{it}^2$ | - | −0.0041 **(−2.49) | - | - | - |
| $L\_outflow_{it}$ | - | - | 0.0118 **(2.20) | - | - |
| $L\_outflow_{it}^2$ | - | - | −0.0007 **(−2.21) | - | - |
| $TEC_{it}$ | 0.0007 *(1.92) | 0.0005 *(1.81) | 0.0005 *(1.73) | −0.1152 **(−2.37) | −0.4122 ***(−2.85) |
| $PRO_{it}$ | −0.0616 *(−1.79) | −0.0514 **(−2.19) | −0.0521 **(−1.98) | 0.2013 *(1.71) | 0.0049 *(1.92) |
| $LQ_{it}$ | 0.2132 ***(7.11) | 0.2063 ***(6.87) | 0.2063 ***(6.86) | 0.5375 **(2.33) | 2.0510 ***(2.99) |
| $EST_{it}$ | −0.1252 ***(−3.71) | −0.1169 ***(−3.46) | −0.1300 ***(−3.86) | 0.6411 **(2.28) | 0.1321 *(1.77) |
| $GDPP_{it}$ | 0.0001 ***(4.57) | 0.0001 ***(4.55) | 0.0001 ***(4.65) | −0.0001(−4.14) | −0.0001 **(−2.36) |
| $FDI_{it}$ | −0.0020 *(−1.69) | −0.0022 *(−1.88) | −0.0019 *(−1.73) | 0.0118 *(1.86) | 0.1015 *(1.80) |
| $WAG_{it}$ | −0.0001 *(−1.81) | −0.0001 *(−1.69) | −0.0001 *(−1.88) | 0.0004 ***(4.92) | 0.0005 **(2.13) |
| $\_Cons$ | 0.5494 ***(5.29) | 0.5107 ***(4.82) | 0.5364 ***(5.12) | −2.8058 ***(−3.04) | −2.2331 (−0.81) |
| Regulatory Inflection Point | Direct Impact | Indirect impact | | - | - |
| | 0.75 | 0.78 | 0.76 | | |
| Maximum or minimum value | 0.62(Max) | 0.58(Min) | 0.61(Min) | - | - |
| N | 540 | 540 | 540 | 540 | 540 |
| F Test | - | - | - | 5.43 *** | 3.07 *** |
| Wald Test | 216.56 *** | 226.64 *** | 224.41 *** | - | - |
| LR Test | 360.18 *** | 344.13 *** | 361.46 *** | - | - |

Remarks: t statistics in parentheses: * $p < 0.1$, ** $p < 0.05$, *** $p < 0.01$.

**Table 3.** Regression results of the impact of environmental regulations on output of different types of industries.

| Variables | Outputs | | Output Growth Rate | |
|---|---|---|---|---|
| | **Pollution-Intensive Industries** | **Clean Industries** | **Pollution-Intensive Industries** | **Clean Industries** |
| | **(1)** | **(2)** | **(3)** | **(4)** |
| $ENR_{it}$ | 396.7292 ***(7.10) | 493.2701 ***(7.74) | −0.0246 **(−2.46) | −2.0107 *(−1.66) |
| $ENR_{it}^2$ | - | - | - | 0.6948 *(1.70) |
| $TEC_{it}$ | 0.0301 ***(6.16) | 0.0933 ***(16.73) | 0.0718 ***(13.48) | 0.1267 ***(19.33) |
| $PRO_{it}$ | −7698.965 ***(−14.80) | −8481.071 ***(−14.28) | 0.0143 ***(8.27) | 0.0638 ***(6.39) |
| $LQ_{it}$ | 1538.719 **(2.47) | 1587.726 **(2.23) | 0.1049 ***(5.66) | 0.1515 ***(4.58) |
| $EST_{it}$ | 182.3217 (0.59) | 17.4476 (0.05) | 0.0156 *(1.82) | 0.0530 *(1.79) |
| $GDPP_{it}$ | 0.0176 *(1.95) | 0.0246 **(2.38) | 0.0718 **(2.48) | 0.0626 **(2.05) |
| $FDI_{it}$ | −158.7626 ***(−4.56) | −160.6637 ***(−4.05) | −0.0028 ***(−4.62) | −0.0030 ***(−5.71) |
| $WAG_{it}$ | −0.0341 (−0.64) | −0.0286 (−0.47) | −0.0618 (−0.92) | −0.0816 (−1.35) |
| $\_Cons$ | 3794.05 ***(4.42) | 4009.825 ***(4.09) | 0.0484 (0.30) | 1.6831 *(1.86) |
| Time Fixed | YES | YES | YES | YES |
| N | 540 | 540 | 510 | 510 |
| F Test | 73.81 *** | 144.83 *** | 25.46 *** | 14.88 *** |

Remarks: *t* statistics in parentheses: * $p < 0.1$, ** $p < 0.05$, *** $p < 0.01$.

### 4.3. The Impact of Environmental Regulation on GTFP

#### 4.3.1. The Overall Impact of Environmental Regulation on GTFP

Model (1) in Table 2 shows the regression results of the overall effect of environmental regulation on GTFP. The regression equation is valid overall, due to the Wald value being significant at the 1% level. The results of the LR test of the equation reject the original hypothesis of fixed effects at the 1% level of significance, indicating that it is reasonable to use a panel Tobit model with random effects. The coefficients of both the primary and secondary terms of environmental regulation in model (1) are significant, indicating that there is a significant nonlinear relationship between environmental regulation and industrial GTFP. Specifically, the coefficients of the primary term of environmental regulation are significantly negative and the coefficients of the secondary term are significantly positive, indicating that as the intensity of environmental regulation increases, industrial GTFP increases and then decreases, showing an inverted "U"-shaped relationship.

#### 4.3.2. Indirect Effects of Environmental Regulations on Industrial GTFP-Mediating Role of Inter-Industry Factor Allocation

In models (2) and (3) of Table 2, the coefficients of the primary term and the quadratic term of the allocation of capital and labor among industries are significantly positive and negative, indicating that the industrial GTFP decreases and then increases with the allocation of factors to clean industries (as shown by the decrease in the value), showing a "U"-shaped trend. Combining the results of model (4) and model (5) in Table 2, it is clear that the effect of environmental regulation on industrial GTFP stems from the relative changes in GTFP of pollution-intensive industries and GTFP of clean industries. As the intensity of environmental regulation increases, the allocation ratio of factors in pollution-intensive industries and clean industries decreases, resulting in a "U"-shaped trend of industrial GTFP. Hypothesis 4 is verified.

Models (2) and (3) show that the direct effect of environmental regulation on industrial GTFP has an inverted "U" shape. Comparing the direct and indirect effects of environmental regulation on industrial GTFP, we find that increasing the intensity of environmental regulation oriented to changing the allocation ratio of factors among industries can "reverse" the effect of high-intensity regulation to suppress industrial GTFP and achieve the effectiveness of government environmental management instruments. Comparing the magnitude of the inflection points in models (1), (2), and (3), we find that the inflection points in models (2) and (3) are larger than those in model (1), indicating that ignoring the mediating role of the inter-industry factor allocation ratio overestimates the inflection point of environmental regulation. The change of factor allocation ratio among industries can improve the decline of industrial GTFP due to the high intensity of regulation, to a certain extent.

### 4.4. Robustness Tests

In order to ensure the reliability and scientific validity of the regression results, this paper uses three methods to test the robustness of the results. First, different proxy variables for the core variables will cause differences in the results, so this paper replaces the proxy variables for environmental regulation with unit pollution control investment to re-regress. Second, outliers can cause errors in the estimation results, and the truncated and shrunken tails of each variable can precisely solve the bias caused by outliers effectively. In this paper, a 1% two-way truncation and a 1% two-way tail reduction are applied to all variables. Third, the ordinary least squares (OLS) method was used to regress the original equations. The results of the robustness test show that the relationship and significance between environmental regulation, inter-industry factor input structure, and industrial GTFP are consistent with the results of the benchmark regression after replacing the core explanatory variables, applying tailoring and truncation to all variables and changing the regression method. Therefore, the existing results in this paper have strong robustness. Due to space limitations, the results of the robustness test are shown in Appendix A.

### 4.5. Endogenous Issues

The endogenous factors between variables are not taken into account in the above analysis. For example, areas with low industrial GTFP are relatively more polluted and tend to use higher intensity environmental regulations in order to reduce pollution, which leads to an inverse causal relationship between environmental regulations and industrial GTFP. The existence of endogenous factors can bias the estimation results. In this paper, the endogenous factors are treated by using the air flow coefficient as an instrumental variable for environmental regulations.

On the one hand, the more air mobility a region has, the less pollution it has, and the larger the composite index of the proxy variable for environmental regulation the less pollution it has. Therefore, the air mobility coefficient is correlated with environmental regulation and satisfies the hypothesis of correlation of instrumental variables [31]. On the other hand, the air flow coefficient is used to measure the geographical characteristics of a region, satisfying the endogenous assumption of the instrumental variable, while the industrial GTFP is used to measure its economic characteristics [32]. Drawing on the study of Shiyi Chen and Dengke Chen [33], this paper uses the air flow coefficient to measure environmental regulation. The construction method is as follows:

$$VC_{it} = WS_{it} \times BLH_{it}, \tag{31}$$

where $VC_{it}$ is the air flow coefficient, $WS_{it}$ is the wind speed, and $BLH_{it}$ is the atmospheric boundary layer height. The raw data of $WS_{it}$ and $BLH_{it}$ are obtained from the monthly average data of ERA-Interim released by the European Centre for Medium-Range Weather Forecasts (ECMWF). Considering that the data are global data under each latitude and longitude, in order to obtain the data for 30 Chinese provinces from 2000–2017, this paper uses Python to correspond them on the map of China and averages the data under the latitude and longitude in each province.

Table 4 shows the results of the two-step estimation of the system two-stage least squares (2SLS). The F-values of the first-stage regressions are all greater than 10 and significant at the 1% level, indicating that the selected air mobility coefficients do not have weak instrumental variable problems. The results of the second stage indicate that the direct effect of environmental regulation on industrial GTFP is an inverted "U" shape, but the greater allocation of capital and labor factors to clean industries makes industrial GTFP "U"-shaped. Therefore, the shift in the allocation of factors among industries can "reverse" the suppression of industrial GTFP by high-intensity environmental regulations.

**Table 4.** The 2SLS regression results for the effect of environmental regulation on GTFP.

| Variables | First Stage | | Second Stage | First Stage | | Second Stage |
|---|---|---|---|---|---|---|
| | **Allocation of Capital among Industries** | | | **Allocation of Labor among Industries** | | |
| | $ENR_{it}$ | $ENR_{it}^2$ | $GTFP_{it}$ | $ENR_{it}$ | $ENR_{it}^2$ | $GTFP_{it}$ |
| | **(1)** | **(2)** | **(3)** | **(4)** | **(5)** | **(6)** |
| $ENR\_iv_{it}$ | 0.0192 *** (2.84) | 0.1111 *** (3.20) | - | 0.0204 *** (2.95) | 0.0754 * (1.91) | - |
| $ENR\_iv_{it}^2$ | −0.0001 *** (−3.14) | −0.0005 *** (−3.40) | - | −0.0001 *** (−3.25) | −0.0003 ** (−2.30) | - |
| $K\_outflow_{it}$ | 0.5651 ** (2.32) | 3.9412 ** (2.31) | 0.0285 * (1.79) | - | - | - |
| $K\_outflow_{it}^2$ | −0.1227 ** (−1.99) | −0.8618 ** (−2.17) | −0.0034 * (−1.88) | - | - | - |
| $L\_outflow_{it}$ | - | - | - | 0.7677 * (1.81) | 9.4834 *** (2.60) | 0.2132 * (1.85) |
| $L\_outflow_{it}^2$ | - | - | - | −0.2310 * (−1.70) | −2.9166 ** (−2.29) | −0.0634 * (−1.92) |

**Table 4.** *Cont.*

| Variables | First Stage | | Second Stage | First Stage | | Second Stage |
|---|---|---|---|---|---|---|
| | Allocation of Capital among Industries | | | Allocation of Labor among Industries | | |
| | $ENR_{it}$ | $ENR_{it}^2$ | $GTFP_{it}$ | $ENR_{it}$ | $ENR_{it}^2$ | $GTFP_{it}$ |
| | (1) | (2) | (3) | (4) | (5) | (6) |
| $ENR_{it}$ | - | - | 0.3944 *** (2.67) | - | - | 0.3948 ** (2.47) |
| $ENR_{it}^2$ | - | - | −0.0575 * (−1.83) | - | - | −0.0538 * (−1.73) |
| _Cons | YES | YES | YES | YES | YES | YES |
| Control variables | YES | YES | YES | YES | YES | YES |
| N | 540 | 540 | 540 | 540 | 540 | 540 |
| Wald Test | - | - | 113.39 *** | - | - | 129.09 *** |
| Adjust $R^2$ | 0.2387 | 0.1778 | - | 0.2360 | 0.1816 | - |
| F-value of First stage | 8.66 *** | 4.69 *** | - | 9.02 *** | 4.61 *** | - |

Remarks: *t* statistics in parentheses: * $p < 0.1$, ** $p < 0.05$, *** $p < 0.01$.

## 5. The Influence of Heterogeneous Factors

### 5.1. Changes in the Ratio of Capital to Labor within Each Industry at the Time of Factor Input

It is known from the results in Chapter 4 that an increase in the intensity of environmental regulation causes more capital and labor to be invested in clean industries, but are the capital and labor factor inputs synchronized? Do factor inputs cause a change in the ratio of capital and labor factor inputs within pollution-intensive and clean industries? If the factor input ratio changes, how does this change affect the GTFP of the industry as a whole? The following section investigates the impact of factor input ratios using the capital and labor ratios within pollution-intensive and clean industries, respectively, replacing the factor allocation ratios across industries in the original model. Table 5 shows that the increase in the intensity of regulation causes the ratio of capital to labor input within the two industries to increase and then decrease. Therefore, the inputs of capital and labor are likewise not synchronized between industries. When the intensity of regulation is weak, manufacturers tend to add more capital elements. As the intensity of regulation gradually increases, manufacturers' preference for labor factors gradually emerges, and the capital–labor ratio within the two industries changes in the process of additional factor inputs by manufacturers. The increase of capital input relative to labor input significantly reduces industrial GTFP within both pollution-intensive and clean industries, indicating that human capital has a stronger role in enhancing GTFP compared to fixed assets. Comparing the rate of change in the ratio of capital and labor inputs between the two industries, the rate of impact of the capital–labor ratio on industrial GTFP within clean industries is greater than that in pollution-intensive industries ( | −0.0009 | > | −0.0003 | ). This is related to the fact that clean industries tend to have a higher level of pollution control.

**Table 5.** Regression results of the effects of environmental regulations, capital, and labor ratios within each industry on industrial GTFP.

| Variables | $K/L\_pol_{it}$ | $K/L\_cle_{it}$ | GTFP | |
|---|---|---|---|---|
| | **(1)** | **(2)** | **(3)** | **(4)** |
| $ENR_{it}$ | 15.3773 ***(3.57) | 1.9821 *(1.69) | 0.1949 ***(2.95) | 0.2002 ***(3.06) |
| $ENR_{it}{}^2$ | −14.5208 ***(−3.44) | −1.2.4782 *(−1.88) | −0.1279 **(−1.97) | −0.1331 **(−2.07) |
| $K/L\_pol_{it}$ | - | - | −0.0003 *(−1.78) | - |
| $K/L\_cle_{it}$ | - | - | - | −0.0009 *(−1.91) |
| Control Variables | YES | YES | YES | YES |
| _Cons | YES | YES | YES | YES |
| Time Fixed | YES | YES | - | - |
| N | 540 | 540 | 540 | 540 |
| F Test | 9.99 *** | 19.63 *** | - | - |
| Wald Test | - | - | 217.13 *** | 216.58 *** |
| LR Test | - | - | 343.47 *** | 314.85 *** |

Remarks: $t$ statistics in parentheses: * $p < 0.1$, ** $p < 0.05$, *** $p < 0.01$.

### 5.2. Differences in the Effects of Capital and Labor Allocation between Industries

Table 6 shows the differences in the impact of environmental regulation on GTFP when capital and labor factors are allocated between industries. The minimum values of factor allocation coefficients, regulatory inflection points, and GTFP are illustrated here for comparison. First, the absolute values of both the primary and quadratic term coefficients of the inter-industry allocation of capital factors are larger than those of the inter-industry allocation of labor factors, indicating that the inter-industry allocation of capital factors has a stronger impact on GTFP. This is because China's current industrial enterprises are mainly capital-intensive, and such enterprises are more sensitive to changes in capital. Second, comparing the regulatory inflection points of the indirect effects of environmental regulation on GTFP mediated by the inter-industry allocation of capital and labor factors, it can be found that the regulatory inflection point of the inter-industry allocation of labor is located to the right of capital, and the corresponding minimum value of GTFP is larger than that of capital, which shows that the interval of regulation to enhance GTFP is longer and the starting point is higher. That is, the effect of labor inter-industry allocation on the effect of regulation is more persistent. This suggests that, compared to physical capital, human capital is the core driver of high-quality economic development.

**Table 6.** Differences between capital and labor allocation in the indirect effects of environmental regulation on GTFP.

| | | Capital | Labor |
|---|---|---|---|
| Coefficients of factor allocation | Primary term coefficients | 0.0283 ** | 0.0118 ** |
| | Quadratic term coefficients | −0.0041 ** | −0.0007 ** |
| Regulatory Inflection Point | | 0.78 | 0.76 |
| Minimum value of GTFP | | 0.58 | 0.61 |

Remarks: $t$ statistics in parentheses: ** $p < 0.05$.

## 6. Conclusions

In this paper, we divided industries into pollution-intensive and clean industries to construct a two-sector vendor production function with endogenous technological progress. This paper analyzed the effects of environmental regulations on factor inputs and outputs in different industries and derived the mediating role of inter-industry allocation of factors in the impact on industrial GTFP. Then, we used industrial panel data of 30 Chinese provinces from 2000–2017 and an EBM-ML model to measure the whole industrial GTFP and used Tobit panel regression and instrumental variables to test the effect of environmental regulation on industrial GTFP and the role of inter-industry factor allocation ratio on

the effect of regulation. The air flow coefficient was then used as an instrumental variable of environmental regulation to solve the problem of endogenous factors in the original model. Finally, the results were discussed with respect to the heterogeneity of factors. The paper draws the following conclusions:

1.  The relative magnitude of output compensation and innovation compensation in the production and R&D sectors changes the impact of environmental regulation on factor inputs and outputs across industries, and the combined effect of both inputs and outputs affects the level of GTFP. From the perspective of factor inputs, as the intensity of regulation increases, factors are allocated more toward clean industries. From the perspective of output, the pollution-intensive industry decreases the output growth rate with the increase of regulation intensity because the output compensation is greater than the innovation compensation, and the clean industry increases the output growth rate with the increase of regulation intensity because the innovation compensation is greater than the output compensation. Therefore, the government should further improve the innovation incentive policy for enterprises and reduce or waive part of the taxes or implement innovation subsidies for industrial enterprises to help them reduce innovation costs and smoothly pass through the technology development period.

2.  The change in the factor allocation ratio among industries can "reverse" the inhibitory effect of high-intensity environmental regulations on industrial GTFP and effectively increase industrial GTFP. In terms of direct effects, the impact of regulation on industrial GTFP is an inverted "U" shape. In terms of indirect effects, the increase of environmental regulations will lead to a greater allocation of capital and labor factors to clean industries, and the industrial GTFP will be in a "U" shape. Therefore, environmental regulation policies should be formulated to promote the change of the factor allocation ratio among industries, so as to promote the reasonable allocation of factor resources among industries, thus effectively alleviating the production pressure brought by high regulation intensity and improving industrial GTFP.

3.  Manufacturers' preference for capital and labor factors varies with the intensity of regulation. When regulation is weak, firms tend to add more capital factors, and as the intensity of regulation increases, firms tend to add more capital factors. It suggests that human capital is stickier to the intensity of regulation. Although the inter-industry allocation of capital factors has a greater intensity on GTFP, the inter-industry allocation of labor has a more lasting effect on the effect of regulation, and human capital is the core driver of high-quality economic development. Therefore, to achieve high-quality development of the industrial economy, the investment in human capital should be increased and the level of technological innovation should be enhanced.

The conclusions of this manuscript argue for the implementation of existing environmental regulation policies and point the way to the development of environmental policies and regulations for the Chinese government. It also shows what kind of environmental regulations can be formulated to promote economic growth while reducing pollution. This manuscript is devoted to the study of environmental regulations that decouple environmental pollution from economic development. On the one hand, it monitors the effects of policy implementation. On the other hand, it provides feasible suggestions for policy formulation.

There are two limitations of this paper: (1) The findings suggest that the relative size of output compensation and innovation compensation affects the ratio of environmental regulation to factor allocation among industries, and the specific values of output compensation and innovation compensation are not calculated here. Future research can use feasible methods to calculate the size of output compensation and innovation compensation and further analyze the impact of the difference between the two types of compensation on the ratio of factor allocation among industries. (2) Restricted by data availability, the research

interval of this paper is 2000–2017. If the data can be updated to 2020, the relevance of the study will be more significant.

**Author Contributions:** Conceptualization, J.Y. and D.Z.; writing—original draft preparation, J.Y.; writing—review and editing, J.Y. and D.Z.; resources, D.Z.; methodology, J.Y.; data curation, J.Y. All authors have read and agreed to the published version of the manuscript.

**Funding:** This research received no external funding.

**Institutional Review Board Statement:** Not applicable.

**Informed Consent Statement:** Not applicable.

**Data Availability Statement:** The data presented in this study are available on request from the corresponding author. The data are not publicly available due to they are manually collected, processed and calculated by the author.

**Conflicts of Interest:** The authors declare no conflict of interest.

### Appendix A

The appendix is the result of the robustness test.

**Table A1.** Regression results of the impact of unit pollution control investment on industrial GTFP.

| Variables | GTFP | | | $K\_outflow_{it}$ | $L\_outflow_{it}$ |
|---|---|---|---|---|---|
| | **(1)** | **(2)** | **(3)** | **(4)** | **(5)** |
| $ENR_{it}$ | 0.0777 ***(4.80) | −0.0353 **(−2.12) | 0.0002 ***(−2.56) | −0.0013 *(−1.67) | −0.0299 *(−1.81) |
| $ENR_{it}^2$ | −0.0784 ***(−2.75) | −0.0054 ***(3.28) | $1.03 \times 10^{-6}$ ***(4.37) | - | - |
| $K\_outflow_{it}$ | - | 0.0025 *(1.89) | - | - | - |
| $K\_outflow_{it}^2$ | - | −0.0001 *(−1.78) | - | - | - |
| $L\_outflow_{it}$ | - | - | 0.0020 **(2.41) | - | - |
| $L\_outflow_{it}^2$ | - | - | −0.0001 **(−2.02) | - | - |
| $TEC_{it}$ | 0.0059 ***(5.85) | $2.39 \times 10^{-6}$ ***(5.89) | $2.28 \times 10^{-6}$ ***(5.67) | −0.1152 **(−2.37) | −0.4122 ***(−2.85) |
| $PRO_{it}$ | −0.5549 ***(−17.91) | −0.5504 ***(−17.39) | −0.5493 ***(−17.80) | 0.2013 *(1.71) | 0.0049 *(1.92) |
| $LQ_{it}$ | 0.0959 **(2.55) | 0.0946 **(2.52) | 0.0999 ***(2.68) | 0.5375 **(2.33) | 2.0510 ***(2.99) |
| $EST_{it}$ | −0.0456 **(−2.10) | −0.0450 **(−2.07) | −0.0377 *(−1.73) | 0.6411 **(2.28) | 0.1321 *(1.77) |
| $GDPP_{it}$ | 0.0129 ***(9.20) | $5.13 \times 10^{-6}$ ***(8.99) | $5.04 \times 10^{-6}$ ***(8.97) | −0.0001 (−4.14) | −0.0001 **(−2.36) |
| $FDI_{it}$ | −0.0001 ***(−3.79) | −0.0001 ***(−3.75) | −0.0001 ***(−3.70) | 0.0118 *(1.86) | 0.1015 *(1.80) |
| $WAG_{it}$ | −0.0076 ***(−3.65) | −0.0074 ***(−3.52) | −0.0069 ***(−3.33) | 0.0004 ***(4.92) | 0.0005 **(2.13) |
| _Cons | 0.7415 ***(12.55) | 0.7460 ***(12.57) | 0.7582 ***(12.73) | −2.8058 ***(−3.04) | −2.2331 (−0.81) |
| N | 540 | 540 | 540 | 540 | 540 |
| F Test | - | - | - | 5.43 *** | 3.07 *** |
| Wald Test | 192.41 *** | 201.91 *** | 207.63 *** | - | - |
| LR Test | 342.67 *** | 345.15 *** | 353.44 *** | - | - |

Remarks: *t* statistics in parentheses: * $p < 0.1$, ** $p < 0.05$, *** $p < 0.01$.

**Table A2.** Regression results after shrinking and censoring.

| Variables | GTFP | | | $K\_outflow_{it}$ | $L\_outflow_{it}$ |
|---|---|---|---|---|---|
| | **(1)** | **(2)** | **(3)** | **(4)** | **(5)** |
| | **1% two-way tail reduction of all variables** | | | | |
| $ENR_{it}$ | 0.2034 ***(3.08) | 0.0447 ***(4.22) | 0.0463 ***(4.42) | −0.2184 **(−2.12) | −0.1263 *(−1.79) |
| $ENR_{it}^2$ | −0.1316 **(−2.03) | −0.0102 ***(−2.58) | −0.0186 ***(−2.76) | - | - |
| $K\_outflow_{it}$ | - | 0.2261 *(3.43) | - | - | - |
| $K\_outflow_{it}^2$ | - | −0.1592 **(−2.45) | - | - | - |
| $L\_outflow_{it}$ | - | - | 0.2205 ***(3.38) | - | - |
| $L\_outflow_{it}^2$ | - | - | −0.1444 **(−2.26) | - | - |
| _Cons | YES | YES | YES | YES | YES |
| Control Variables | YES | YES | YES | YES | YES |
| N | 540 | 540 | 540 | 540 | 540 |
| F Test | - | - | - | 2.95 *** | 3.15 *** |
| Wald Test | 201.41 *** | 213.21 *** | 221.54 *** | - | - |
| LR Test | 349.14 *** | 355.05 *** | 363.21 *** | - | - |

**Table A2.** *Cont.*

| Variables | GTFP | | | $K\_outflow_{it}$ | $L\_outflow_{it}$ |
|---|---|---|---|---|---|
| | (1) | (2) | (3) | (4) | (5) |
| **1% two-way truncation of all variables** | | | | | |
| $ENR_{it}$ | 0.1067 ***(4.39) | 0.0337 ***(3.23) | 0.0353 ***(3.46) | −0.2323 **(−2.19) | −0.1349 *(−1.87) |
| $ENR_{it}^2$ | −0.0569 *(1.83) | −0.0024 *(−1.71) | −0.0026 *(−1.91) | - | - |
| $K\_outflow_{it}$ | - | 0.1261 **(1.97) | - | - | - |
| $K\_outflow_{it}^2$ | - | - | - | - | - |
| $L\_outflow_{it}$ | - | - | 0.1155 *(1.83) | - | - |
| $L\_outflow_{it}^2$ | - | - | −0.0609 *(−1.88) | - | - |
| _Cons | YES | YES | YES | YES | YES |
| Control Variables | YES | YES | YES | YES | YES |
| N | 528 | 528 | 528 | 528 | 528 |
| F Test | - | - | - | 2.98 *** | 3.25 *** |
| Wald Test | 176.32 *** | 184.78 *** | 204.39 *** | - | - |
| LR Test | 365.75 *** | 370.42 *** | 385.33 *** | - | - |

Remarks: *t* statistics in parentheses: * $p < 0.1$, ** $p < 0.05$, *** $p < 0.01$.

**Table A3.** OLS regression results of the fixed effects of environmental regulations on industrial GTFP.

| Variables | GTFP | | | $K\_outflow_{it}$ | $L\_outflow_{it}$ |
|---|---|---|---|---|---|
| | (1) | (2) | (3) | (4) | (5) |
| $ENR_{it}$ | 0.2071 ***(3.22) | 0.0379 **(2.32) | 0.0399 ***(4.33) | −0.0013 *(−1.67) | −0.0299 *(−1.81) |
| $ENR_{it}^2$ | −0.1416 **(−2.25) | −0.0054 **(−2.30) | −0.0029 **(−2.54) | - | - |
| $K\_outflow_{it}$ | - | 0.2320 ***(−3.60) | - | - | - |
| $K\_outflow_{it}^2$ | - | −0.1717 ***(−2.71) | - | - | - |
| $L\_outflow_{it}$ | - | - | 0.2198 ***(3.46) | - | - |
| $L\_outflow_{it}^2$ | - | - | −0.1497 **(−2.41) | - | - |
| $TEC_{it}$ | $1.22 \times 10^{-6}$ ***(4.18) | $1.23 \times 10^{-6}$ ***(4.20) | $1.17 \times 10^{-6}$ ***(4.02) | −0.1152 **(−2.37) | −0.4122 ***(−2.85) |
| $PRO_{it}$ | −0.5495 ***(−18.29) | −0.5467 ***(−17.82) | −0.5415 ***(−18.15) | 0.2013 *(1.71) | 0.0049 *(1.92) |
| $LQ_{it}$ | 0.0964 ***(2.68) | 0.0955 ***(2.65) | 0.1003 ***(2.82) | 0.5375 **(2.33) | 2.0510 ***(2.99) |
| $EST_{it}$ | −0.0860 ***(−4.80) | −0.0855 ***(−4.75) | −0.0709 ***(−3.87) | 0.6411 **(2.28) | 0.1321 *(1.77) |
| $GDPP_{it}$ | $6 \times 10^{-6}$ ***(11.28) | $5.96 \times 10^{-6}$ ***(11.06) | $5.7 \times 10^{-6}$ ***(10.67) | −0.0001(−4.14) | −0.0001 **(−2.36) |
| $WAG_{it}$ | −0.0001 ***(−6.01) | −0.0001 ***(−5.95) | −0.0001 ***(−5.81) | 0.0118 *(1.86) | 0.1015 *(1.80) |
| $FDI_{it}$ | −0.0086 ***(−4.30) | −0.0085 ***(−4.20) | −0.0077 ***(−3.82) | 0.0004 ***(4.92) | 0.0005 **(2.13) |
| _Cons | 0.7855 ***(15.88) | 0.7882 ***(15.81) | 0.8048 ***(16.32) | −2.8058 ***(−3.04) | −2.2331 (−0.81) |
| N | 540 | 540 | 540 | 540 | 540 |
| F Test | 27.1 *** | 27.60 *** | 28.40 *** | 5.43 *** | 3.07 *** |

Remarks: *t* statistics in parentheses: * $p < 0.1$, ** $p < 0.05$, *** $p < 0.01$.

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
