# Peer review of "Research on the Impact of Environmental Regulations on Industrial Green Total Factor Productivity: Perspectives on the Changes in the Allocation Ratio of Factors among Different Industries"

_sustainability, doi:10.3390/su132312947_

Round 1

Reviewer 1 Report

The paper entitled” Research on the Impact of Environmental Regulations on Industrial Green Total Factor Productivity: Perspectives on the changes in the allocation ratio of factors among different industries” deals with actual and interesting topic.
However, I have the following comments that hopefully help the authors
improve their paper:
A research method diagram would be helpful.  This will
provide a snapshot of the research steps followed and a clearer understanding of the paper.
The authors should convince that this contribution is  important. These issues deserve a deeper discussion: 
What are the research results implications for theory and practice?
What are the limitations of the study in terms of the proposed method,
data used, approaches, and/or analysis? How can these limitations be
addressed in future research?
I would suggest  to the authors to improve this work further along those lines.

Reviewer 2 Report

The paper is appropriate to the purpose of the journal.

The paper analyzes the impact of environmental regulations on the input and output factor of different industries with the help of a two-sector manufacturer model of endogenous technological progress.

The validation in practice of the model is done with the help of data collected from 30 provinces in China's industry from 2000 to 2017.

In writing the paper it is preferable that the acronyms be explained at the first meeting in the text, eg GTFP even if it appears in the title of the paper, GDP, EBM-ML Model etc.

I appreciate that it should better explain why Cobb-Douglas production functions are used. 

It should also explain the profit of a manufacturer equation context, especially what the term Y means, which does not appear anywhere explained.

On page 8 the phrase „Take the environmental regulation level when the output。” has no clear meaning.

A bibliographic reference to MaxDEA would be welcome.

It would be good to explain the reason on which the composite indicators were constructed as proxies for environmental regulation.

Overall, the paper is well structured and meets the requirements of a research article.

Reviewer 3 Report

The present study deals with a very interesting and fascinating topic. However, despite the potential relevance of its contribution, I believe that it presents some issues that need to be tackled.
In the following, you can find some suggestions and comments that I hope may be useful to improve the contribution of your study.
First, I believe the authors should deeper discuss the theoretical relevance of their study, clearly describing why it is relevant.
Second, I suggest the authors to further discuss the role of environmental innovations (Aldieri et al., 2021; Ardito et al., 2019).
Third, the choice of setting and data need to be justified.
Fourth, the paper may benefit from a professional copy editing.

References

Aldieri, L., Brahmi, M., Chen, J., Vinci, C. P. (2021). Knowledge Spillovers and Technical Efficiency for Cleaner Production: An Economic Analysis from Agriculture Innovation. Journal of Cleaner Production, https://doi.org/10.1016/j.jclepro.2021.128830.

Ardito, L., Messeni Petruzzelli, A., Peruffo, E., Pascucci, F. (2019). Inter-firm R&D collaborations and green innovation value: The role of family firms' involvement and the moderating effects of proximity dimensions. Business Strategy and the Environment, Vol. 28, pp. 185-197.

Round 2

Reviewer 3 Report

The paper has been improved according to the reviewers 'comments.  Now the manuscript can be accepted for publication.